# The association between nurse staffing levels and the timeliness of vital signs monitoring: a retrospective observational study in the UK

Oliver C Redfern,[1] Peter Griffiths ![ORCID],[2] Antonello Maruotti,[3] Alejandra Recio Saucedo,[2] Gary B Smith ![ORCID],[4] The Missed Care Study Group

[1]Nuffield Department of Clinical Neurosciences, Oxford University, Oxford, UK
[2]NIHR Collaboration for Leadership in Applied Heath Research and Care (Wessex), University of Southampton, Southampton, UK
[3]Dipartimento di Scienze Economiche, Libera Universita Maria Santissima Assunta, Roma, Italy
[4]School of Health and Social Care, University of Bournemouth, Bournemouth, UK

**Correspondence to**
Professor Peter Griffiths;
peter.griffiths@soton.ac.uk

## ABSTRACT

**Objectives** Omissions and delays in delivering nursing care are widely reported consequences of staffing shortages, with potentially serious impacts on patients. However, studies so far have relied almost exclusively on nurse self-reporting. Monitoring vital signs is a key part of nursing work and electronic recording provides an opportunity to objectively measure delays in care. This study aimed to determine the association between registered nurse (RN) and nursing assistant (NA) staffing levels and adherence to a vital signs monitoring protocol.

**Design** Retrospective observational study.

**Setting** 32 medical and surgical wards in an acute general hospital in England.

**Participants** 538 238 nursing shifts taken over 30 982 ward days.

**Primary and secondary outcome measures** Vital signs observations were scheduled according to a protocol based on the National Early Warning Score (NEWS). The primary outcome was the daily rate of missed vital signs (overdue by ≥67% of the expected time to next observation). The secondary outcome was the daily rate of late vital signs observations (overdue by ≥33%). We undertook subgroup analysis by stratifying observations into low, medium and high acuity using NEWS.

**Results** Late and missed observations were frequent, particularly in high acuity patients (median=44%). Higher levels of RN staffing, measured in hours per patient per day (HPPD), were associated with a lower rate of missed observations in all (IRR 0.983, 95% CI 0.979 to 0.987) and high acuity patients (0.982, 95% CI 0.972 to 0.992). However, levels of NA staffing were only associated with the daily rate (0.954, CI 0.949 to 0.958) of all missed observations.

**Conclusions** Adherence to vital signs monitoring protocols is sensitive to levels of nurse and NA staffing, although high acuity observations appeared unaffected by levels of NAs. We demonstrate that objectively measured omissions in care are related to nurse staffing levels, although the absolute effects are small.

**Study registration** The data and analyses presented here were part of the larger Missed Care study (ISRCTN registration: 17930973).

## Strengths and limitations of this study

► This was a large retrospective cohort study exploring the association between levels of nursing staff in an acute hospital and adherence to a vital signs monitoring protocol.
► In contrast to previous studies that relied on nursing staff self-reporting missed care, we used an objective measure derived from electronically recorded vital signs.
► This study was limited to a single hospital and we were only able to partially adjust for other factors that might affect protocol adherence.

## INTRODUCTION

Reports from around the world have highlighted poor nursing care as a cause of avoidable harm.[1–3] Perhaps unsurprisingly, there is mounting evidence that quality of care deteriorates when wards are understaffed,[4] yet the extent to which low staffing leads directly to worse outcomes for patients remains in dispute.[5] A number of studies have explored whether nursing work that is delayed or left incomplete (often referred to as 'missed care'[6]) provides a plausible causal mechanism leading to worse patient outcomes, as nurses do not have the capacity to deliver all required care when staffing levels are inadequate.[4 7] However, details of nursing activities are not always routinely collected or recorded in standard formats, or in systems that can be easily interrogated, by healthcare providers. Therefore, it is difficult to measure the timing of care or the extent to which care is delivered.[8] Consequently, the evidence supporting an association between missed care and staffing levels is largely based on nurses' self-reports.[4 9 10]

Recording patients' vital signs is a fundamental aspect of nursing work, and a key component of patient surveillance:

infrequent monitoring can cause signs of clinical deterioration to be missed, leading to delays in administering remedial treatment.[3 11 12] A Europe-wide cross-sectional study (RN4CAST - registered nurse forecasting study) found that 27% of nursing staff reported missing at least some necessary patient surveillance on their last shift.[9] The failure to properly observe and record vital signs observations has been noted as a factor in inquiries into the cause of preventable death in hospital patients.[13]

In response to the increasing recognition that monitoring vital signs is suboptimal, a number of protocols which define observation schedules have been developed and implemented. For example, on general medical and surgical wards, UK guidelines recommend that the frequency of monitoring is directed by the National Early Warning Score (NEWS).[12] This is a score that provides a composite measure of patients' physiological abnormalities, based on vital signs measurements: in general, the higher the score, the more frequently patients should be observed. Internationally, a range of similar early warning or escalation systems are used to guide the observation and escalation of care for at-risk patients.[14] However, retrospective studies have shown, at best, partial adherence to monitoring protocols, particularly at night.[15 16]

Inadequate staffing is one possible explanation for this lack of adherence, as it may reduce nurses' capacity to monitor and intervene to prevent deterioration. This could be one explanation for the association between low nurse staffing levels and increased mortality, which has been demonstrated in many studies worldwide.[5] Yet, existing studies linking low staffing to missed care have exclusively used self-report by nurses derived from cross-sectional surveys.[4] Such studies suffer a number of limitations, including common-method bias, because all variables are derived from the same self-report survey.[17]

In this retrospective observational study of an acute hospital in England, we used routinely collected records of vital signs and other clinical and administrative data, including the electronic rostering database, to investigate whether adherence to the hospital's vital signs monitoring protocol was sensitive to the daily levels of nursing staff.

## METHODS

### Study design and setting

This was a retrospective longitudinal observational study of 32 wards in a large acute hospital in the south of England over 3 years (April 2012 to March 2015).

### Data sources and linkage

#### Sources

This study combined data from four sources. Information on patients (admissions, ward transfers) was obtained from the patient administration system (PAS), allowing us to calculate bed occupancy and the number of admissions to each ward. Vital signs observations were obtained from the Vitalpac system.[18] Data items were: anonymised admission identifier, time of observation, NEWS,[12] time to next observation.

Levels of nurse staffing were derived from two source databases. For standard contractual shifts, we extracted data from an electronic rostering system, detailing the date, location, number of hours and grade of each nurse for every shift. The second source was a similar database recording all bank (extra contractual work by staff employed by the hospital) and agency (staff employed through an external agency) shifts.

In total we identified 538 238 shifts worked over the study period by either registered nurses (RNs; fully qualified nurses on the Nursing and Midwifery Council Register with university diploma or degree level qualification or equivalent) or NAs (nursing assistant personnel with no formal training requirements or registration, typically employed in roles described as healthcare assistants in National Health Service pay bands 2–4). We did not have access to data on shifts undertaken by student nurses. However, they are considered supernumerary for the purposes of staff allocation.

#### Linkage

Nursing shifts worked on each day of the study were linked to vital signs observations and admission data (from PAS) using ward location identifiers and time stamps. For each of the 32 wards, we calculated daily patient and staffing levels. From a theoretical maximum of 35 040 ward days (365 days × 3 years × 32 wards) there were 1822 (5.2%) ward days where one or more of the study wards was closed and 2236 (6.4%) ward days where we could not properly match patient records and staffing. This was generally when wards relocated and nursing shifts were still recorded in the old location for a while. Additionally, periods when wards opened, closed or transferred were often associated with unusual values for patient numbers or staff/patient ratios due to low patient census or delayed recording of staff transfers to the new unit. Therefore, we excluded all ward days where the patient census fell below 25% of the ward median. We were unable to link e-roster to the staff taking the observations as no standard identifier was available although we attempted to identify the grade of staff taking the observations using a descriptive field in the Vitalpac system.

#### Outcomes

A total of 2864 975 complete sets of vital signs were available for analysis. The primary outcome of the study was *missed* vital signs observations. The secondary outcome was *delayed* observations.

Both outcomes were calculated with reference to the hospital's vital signs monitoring protocol. The protocol is based on NEWS,[12] where the level of derangement in vital signs (the NEWS value) is aggregated into a single integer. This is then used to determine when the patient should next be observed—in general, higher scores prompt more frequent observation. For example, if the NEWS value is 2, the patient should be observed at least

every 6 hours. Patients with the lowest score (NEWS=0) are observed every 12 hours and those with higher scores more frequently.

We defined a vital signs observation as *missed* if overdue by more than 67% of the expected time to next observation determined by the previous NEWS value. Similarly, an observation was *delayed* if overdue by more than 33% of the expected time to next observation determined by the previous NEWS value. For example, if the next observation was due in 60 min, it was classified as *delayed* if taken >80 min after the previous observation and *missed* if taken >100 min later.

For subgroup analyses, missed/delayed observations were further stratified in acuity categories according to the previous NEWS value. The study hospital's monitoring protocol (see online supplementary material A1) was used to define the following groups:

► Low: where previous NEWS value <3.
► Medium: where previous NEWS value was between 3 and 5.
► High: where previous NEWS value >5

### Exposures

For each study day on each ward, we calculated the average staffing levels in hours per patient per day (HPPD) for both RNs and NAs. HPPD was calculated by dividing the total number of nursing hours worked by the daily bed occupancy (for that ward). Daily bed occupancy was calculated from the PAS database where a value of 1 indicates a single bed being occupied continuously for 1 day. An HPPD of 24 indicates one-to-one nursing.

To account for variations in other aspects of nursing workload, we derived variables to quantify admission rates ('patient turnover') and the proportion of observations that were for patients requiring 4-hourly or more frequent observation on each day of the study (ie, patient with NEWS value ≥3, 'higher acuity'). Patient turnover was calculated by dividing total daily RN staffing (in days) by the number of new admissions.

### Statistical methods

We chose mixed-effects Poisson regression as our modelling framework to examine the relationship between missed/delayed observations and staffing. Random effects terms were introduced for each ward. All other co-variates were added as fixed effects in the models. Where not otherwise stated, all summary measures are reported using median and IQR. All analyses were undertaken using the R statistical environment V.3.5[19] and mixed-effects models were fit using the lme4 package.[20]

By modelling the effect of each staff group separately, we considered the extent to which the labour inputs from one group might substitute for the other. Additionally, we tested for potential that NAs acted as labour complements, enhancing the effectiveness of RNs by adding interaction terms to each model. We assessed whether these terms improved model fit by examining the Akaike

| Table 1 | Admission characteristics for study participants |
|---|---|
| Admissions, N | 138 133 |
| Emergency admissions, N (%) | 108 865 (79) |
| Elective, N (%) | 29 268 (21) |
| Age median, (range) | 66.6 (16.0–106) |
| Charlson Comorbidity Index, median (range) | 3 (0–98) |
| First NEWS, median (range) | 1 (0–19) |
| Low (NEWS <3) N (%) | 102 674 (74) |
| Medium (NEWS 3–5) n (%) | 27 409 (20) |
| High (NEWS >5) N (%) | 8050 (6) |
| Length of stay in days, median (range) | 2.73 (0.150–933) |
| In-hospital mortality N (%) | 5662 (4.1) |

NEWS, National Early Warning Score.

information criterion (AIC) and Bayesian information criterion (BIC).

### Data de-identification

All readily identifiable information for patients and staff was removed at source. Internal identifiers were anonymised prior to transfer to the research database. Consequently, it was not possible for the research team to identify participants in the study.

### Patient and public involvement

As part of the parent study, we undertook a series of consultations with public, patient and clinical experts/stakeholders (including health services managers and ward-based nurses). These discussions were used to explore views on balancing nursing skill mix (RNs and NAs) on the wards and the factors affecting adherence to current vital signs protocols. An independent lay researcher was also part of the research team and advised on public engagement.

### RESULTS

#### Patient admissions, vital signs and staffing levels

Vital signs observations from 138 133 patient admissions (table 1) were analysed after exclusion criteria were applied (online supplementary figure A2). Patients had a median age of 66.6 years, median Charlson Comorbidity Index of 3 and 79% of were admitted as emergencies. The median length of hospital stay was 2.7 days and the cohort had a 4.1% mortality rate.

On average, 17.1% of all observations across the study wards were classified as missed and 31.3% were delayed. Six per cent of observations were preceded by a high acuity (NEWS >5) score, of which 44% were classified as missed and 53.5% of observations were delayed. Table 2 shows the rate of delayed and missed observations across the 32 study wards. The rate of missed observations varied substantially between wards, with the highest

**Table 2** Percentage of missed and delayed observations for each of the 32 study wards

| Ward | All observations | | High acuity observations | |
|---|---|---|---|---|
| | % Delayed | % Missed | % Delayed | % Missed |
| Surgical: gynaecological | 19.8 | 10.4 | 38.2 | 30.3 |
| Medical: gastroenterology | 42.9 | 25.8 | 61.0 | 52.3 |
| Medical: gardiology/gastroenterology | 45.5 | 24.7 | 62.8 | 52.2 |
| Medical/surgical: cardiac high care | 25.6 | 14.0 | 48.8 | 39.7 |
| Surgical: emergency orthopaedic (spinal) | 33.2 | 19.1 | 46.9 | 37.4 |
| Medical: general | 40.9 | 22.4 | 64.1 | 55.1 |
| Medical: general | 30.8 | 14.4 | 39.8 | 32.0 |
| Surgical: emergency orthopaedic (head injury) | 24.3 | 10.9 | 44.4 | 35.9 |
| Surgical: elective orthopaedic | 21.3 | 11.9 | 29.8 | 23.3 |
| Surgical: older people | 31.9 | 17.4 | 39.7 | 30.7 |
| Surgical: general urology, vascular, plastic | 29.2 | 15.0 | 45.9 | 34.4 |
| Surgical: head and neck | 29.1 | 12.9 | 53.2 | 43.7 |
| Surgical: general, upper Gastro Intestinal | 21.0 | 8.8 | 36.3 | 27.9 |
| Surgical: general/colorectal | 23.5 | 10.6 | 44.1 | 36.2 |
| Medical: respiratory high care and step down | 52.6 | 38.5 | 71.8 | 64.2 |
| Medical: respiratory | 47.7 | 30.7 | 63.5 | 53.4 |
| Rehabilitation: neuro | 61.2 | 45.2 | 56.2 | 47.9 |
| Medical: older people | 28.0 | 15.2 | 51.0 | 41.8 |
| Rehabilitation: stroke (older people) | 52.5 | 35.8 | 53.1 | 44.1 |
| Medical: acute stroke | 40.9 | 19.3 | 58.7 | 49.9 |
| Medical: radiotherapy haematology/oncology | 24.3 | 11.5 | 54.8 | 44.7 |
| Medical: older people | 32.8 | 16.4 | 56.2 | 45.5 |
| Medical: older people | 39.6 | 19.0 | 58.4 | 47.3 |
| Medical: older people | 38.2 | 20.6 | 59.7 | 49.7 |
| Medical: older people | 36.7 | 17.2 | 60.6 | 48.9 |
| Medical/surgical: elective and investigations | 18.7 | 8.8 | 37.1 | 30.5 |
| Medical: renal high care | 25.6 | 13.2 | 45.8 | 36.4 |
| Medical: renal | 21.9 | 10.8 | 46.5 | 38.4 |
| Surgical: renal transplant | 16.3 | 7.6 | 38.3 | 31.5 |
| Medical: emergency admissions | 19.7 | 9.1 | 50.4 | 39.0 |
| Surgical: admissions | 15.4 | 5.6 | 39.1 | 31.7 |
| Surgical: high care | 9.8 | 5.5 | 31.0 | 22.4 |

levels seen on the neurorehabilitation and respiratory high care wards (45% and 39%, respectively). Mean staffing levels for RNs were 4.75 HPPD, with high variation both within and between wards (online supplementary material A3). On average, the within-ward SD of staffing levels was 18% of the mean. Attempts to identify staff groups involved in taking observations were hampered by lack of standard coding. Across all wards an average 15% of observations was recorded as being taken by an NA (16% for low acuity observations, 15% for high acuity observations). However, the lack of standard coding and the large proportion of observations

attributed to 'unknown' staff led us to judge these data as unreliable, and so we did not consider them further in the analysis.

### Relationship between staffing levels and missed observations

To examine the relationship between missed/delayed observations and staffing levels, we first considered *all* observations. We then performed a subgroup analysis, stratifying observations by acuity (low/medium/high, see Methods). Results for the low and medium acuity subgroups are in the online supplementary material (A4).

**Table 3** Mixed-effects Poisson regression: association between staffing and all missed observations with (A) and without (B) inclusion of a linear interaction term between RN and NA staffing levels

| Model | A | | | B | | |
|---|---|---|---|---|---|---|
| | IRR | 95% CI | P value | IRR | 95% CI | P value |
| RN staffing | 0.983 | 0.979 to 0.987 | <0.001 | 0.981 | 0.977 to 0.985 | <0.001 |
| NA staffing | 0.954 | 0.949 to 0.958 | <0.001 | 0.957 | 0.952 to 0.961 | <0.001 |
| Patient turnover | 1.01 | 1.01 to 1.01 | <0.001 | 1.01 | 1.01 to 1.02 | <0.001 |
| Observations in higher acuity patients | 4.83 | 4.68 to 4.99 | <0.001 | 4.8 | 4.65 to 4.96 | <0.001 |
| RN staffing × NA staffing | | | | 1.01 | 1.01 to 1.01 | <0.001 |

Model A: AIC 215974; BIC 216033. Model B: AIC 216062; BIC 216112.
AIC, Akaike information criterion; BIC, Bayesian information criterion; IRR, incidence rate ratio; NA, nursing assistant; RN, registered nurse.

## All observations

Table 3 (Model A) shows the relationship between staffing levels and measure of daily nursing workload with the rate of all missed observations. The rate of missed observations was significantly associated with levels of both RN (p<0.0001) and NA staffing (p<0.001). The magnitude of the effect (incidence rate ratio, IRR) was greater for NAs (IRR 0.954, 0.949–0.958) than for RNs (IRR 0. 983, 0.979–0.987). Measures of admissions per RN and the proportion of higher acuity patients were also highly significant (p<0.001). We introduced a linear interaction term between RN and NA staffing levels into the model, as we hypothesised that levels of one staffing group may be dependent on the effect of the other. Addition of this interaction term (table 3, Model B) was significant (p<0.001) and improved model fit (AIC 215 974 vs 216 062). Similar relationships were observed for the secondary outcome, delayed observations (see online supplementary material A5).

To further explore the relationship between the two nursing groups and the nature of the interaction, we categorised staffing levels into tertiles (online supplementary table A6). The coefficients from this model were used to visualise the effects of various combinations of staff (figure 1). Any additional hours from either staff group reduced the rate of missed observations compared with when staffing from both groups was low. Increasing NA staffing from low to medium was associated with substantial reductions in missed observations for all levels of RN staffing. However, increasing NA staffing from medium to high was only associated with a further reduction in missed observations when RN staffing was low, and even then only by a small amount. Conversely, increasing levels of RN staffing was always associated with a reduction in missed care, regardless of the levels of NA staffing.

## High acuity observations

Table 4 shows equivalent models (Models A and B) in the subgroup of high acuity observations. In this group, only higher levels of RNs were significantly (p<0.001) associated with reductions in the rate of missed observations (IRR 0.982, 95% CI 0.972 to 0.992). Addition of a linear interaction between RN and NA staffing did not alter the size or significance of the relationship between RN staffing and missed observations, and model fit worsened.

## DISCUSSION

### Main findings

This is the first study to examine the relationship between nurse staffing levels and an objective measure of missed care. Furthermore, it is the only study of missed care to focus specifically on vital signs monitoring, which has been implicated in the causal pathway between low staffing and increased mortality.[13] Our results show that higher levels of staffing for both RNs and NAs were associated with significantly lower rates of missed observations. There was significant interaction between the effects of RN and NA staffing levels. Rates of missed or delayed *high* acuity observations were only sensitive to the level of RN staffing with no evidence of interaction between the two staffing levels.

Monitoring vital signs is a fundamental component of the 'Chain of Prevention', a tool that describes the processes required to identify and prevent patient deterioration.[21] Nursing staff clearly play a key role in this

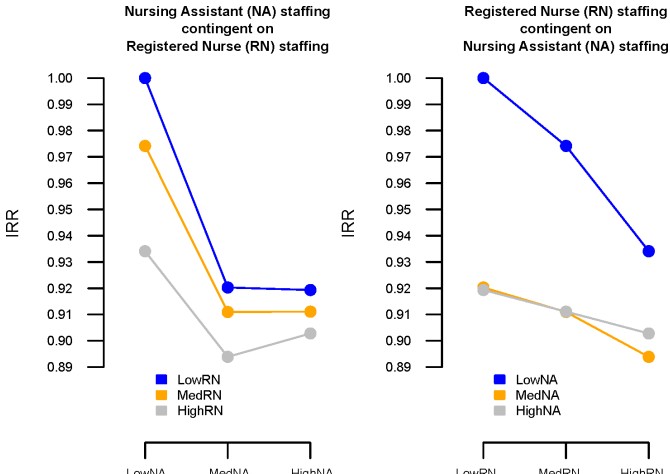

**Figure 1** Partial dependency plots showing interaction effects between levels of registered nurse (RN) and nursing assistant (NA) for all missed observations. IRR, incidence rate ratio.

**Table 4** Mixed-effects Poisson regression: association between staffing and high acuity missed observations with (A) and without (B) inclusion of a linear interaction term between RN and NA staffing levels

| Model | A | | | B | | |
|---|---|---|---|---|---|---|
| | IRR | 95% CI | P value | IRR | 95% CI | P value |
| RN staffing | 0.982 | 0.972 to 0.992 | <0.001 | 0.982 | 0.972 to 0.992 | <0.001 |
| NA staffing | 1 | 0.990 to 1.01 | 0.822 | 1 | 0.991 to 1.01 | 0.791 |
| Patient turnover | 0.997 | 0.988 to 1.01 | 0.59 | 0.997 | 0.988 to 1.01 | 0.582 |
| Observations in higher acuity patients | 1.01 | 0.936 to 1.09 | 0.769 | 1.01 | 0.937 to 1.09 | 0.747 |
| RN staffing × NA staffing | | | | 0.999 | 0.994 to 1.00 | 0.64 |

Model A: AIC 76747; BIC 76796. Model B: AIC 76749; BIC 76806.
AIC, Akaike information criterion; BIC, Bayesian information criterion; IRR, incidence rate ratio; NA, nursing assistant; RN, registered nurse.

process[3] [22] and adherence to monitoring protocols provides a plausible mechanism where 'missed care' could directly lead to adverse outcomes for patients. Our results are consistent with self-reports of nurses[23] and other studies that have highlighted compliance issues with monitoring.[15] [24] [25]

In the face of ongoing shortages of RNs in many countries, NAs and equivalent staff are increasingly deployed to support RNs to undertake some tasks that would otherwise be undertaken by RNs.[26] With regard to the overall rate of missed observations, there is evidence that NAs may act as labour substitutes for RNs in completing observations in a timely fashion. However, this relationship does not apply for higher acuity patients and their observations. The absence of a main effect for NA staffing, in tandem with no significant interaction effect, suggests that NA staff are neither an effective substitute nor a complementary resource (ie, enhancing the ability of RN staffing to deliver observations) for timely observation of acutely unwell patients. This seems a surprising finding, given that a key raison d'être for NAs is to support the work of RNs.[27]

While higher acuity observations are a relatively small proportion of all those taken (6%), patients with a NEWS >5 are at substantially increased risk of dying or experiencing an adverse event such as cardiac arrest within the next 24 hours.[28] Work undertaken by NAs could release RNs to focus on acutely unwell patients (complementarity) but there is some evidence suggesting that NAs are routinely undertaking observations in acutely unwell patients in some settings.[24] Records of the staff group who performed observations in the current study were not fully reliable, although our data are consistent with NAs taking a substantial number of observations for both low acuity and high acuity patients. Absence of substitution and complementarity for NAs in relation to missed nursing care has been demonstrated previously[29] and this finding does serve to emphasise the importance of RNs in ensuring safe care for patients at risk of deterioration. However, the role of the two different staff groups in providing this care merits further investigation.

A key finding of our study is that nurse staffing has a relatively small effect on whether or not vital signs are taken in accordance with protocol. For example, adding one extra hour of RN care per patient per day would result in an absolute reduction of less than 1% in the number of missed high acuity observations. Given the high levels of missed observations (nearly 45% in some wards), it seems clear that most deviations from protocol are attributable to factors other than the number of staff available to make observations. This may be related to the fact that the precise recommended frequencies for monitoring are based on expert opinion[11] [12] and not supported by direct evidence.[30] Consequently, RNs in particular are likely to be exercising their clinical judgement when deciding how often to obtain a set of vital signs. Therefore, we question whether measuring absolute adherence to observation protocols is a valid measure of quality, as it only partially meets the criteria for a good indicator.[31] [32] It might also be that other components of the 'chain of prevention', such as escalating abnormal observations appropriately, are more sensitive to levels of RNs.[21] Nonetheless, reductions in compliance at a ward level may be indicative of deterioration in quality of care and the clinical importance of the small changes we observed are unclear. However, the weak association we observed between staffing levels and compliance with the vital signs observation protocol suggests that the promise that this measure could provide a leading indicator for staffing problems that might lead to poor outcomes may not be realised.[4]

## LIMITATIONS

The main limitation of our study is that it relies on observational data from a single acute hospital. We can therefore only demonstrate an association, rather than causal link, between adherence to monitoring protocols and levels of nursing staff. However, previous studies have relied on cross-sectional designs where levels of missed care and staffing are derived from staff surveys.[23] Our study design eliminated a number of plausible alternative explanations for the association, including common methods bias.

While using routinely collected vitals to quantify adherence to monitoring protocols gives a more objective measure of missed care, it is not without its own

limitations. We were unable to exclude observations from our analysis that were missed for valid clinical or logistic reasons, such as when patients were away from the ward (eg, for radiological or surgical procedures). A previous study in the same hospital also showed that nursing staff are more reluctant to wake patients at night,[33] which could account for some missed observations.

Although we adjusted for daily staffing requirements by incorporating admission rates and the proportion of higher acuity patients into our multilevel models, we were unable to account for other demands on nursing staffing (eg, personal care needs). However, this potential source of bias would tend to underestimate the effect of low staffing, if staffing is increased when demand is high.

## CONCLUSIONS

This is the first study to demonstrate an association between nurse staffing levels and an objective measure of complete and timely care in relation to monitoring patients' vital signs, a key mechanism hypothesised to explain the link between low nurse staffing and increased mortality. Compliance with vital signs monitoring schedules is lower when levels of RN and NA staff are lower, although substantial increases in numbers of staff would be required to effect meaningful increase in adherence. It is likely that other factors, such as clinical judgement, are the main drivers of non-adherence.

**Collaborators** The Missed Care Study Group comprises Peter Griffiths (University of Southampton, Health Sciences, National Institute for Health Research Collaboration for Applied Health Research and Care (Wessex), Karolinska Instutet, Department of Learning, Informatics, Management and Ethics,Portsmouth Hospitals NHS Trust, Clinical Outcomes Research Group), Jane Ball (University of Southampton, Health Sciences, National Institute for Health Research Collaboration for Applied Health Research and Care (Wessex), Karolinska Instutet, Department of Learning, Informatics, Management and Ethics), Karen Bloor (University of York, Health Sciences), Dankmar Böhning (University of Southampton, Health Sciences), Jim Briggs (University of Portsmouth, Centre for Healthcare Modelling and Informatics, Portsmouth Hospitals NHS Trust, Clinical Outcomes Research Group), Chiara Dall'Ora (University of Southampton, Health Sciences, National Institute for Health Research Collaboration for Applied Health Research and Care (Wessex)), Anya De longh (Independent lay researcher), Jeremy Jones (University of Southampton, Health Sciences), Caroline Kovacs (University of Portsmouth, Centre for Healthcare Modelling and Informatics), Antonello Maruotti (University of Southampton, Health Sciences), Paul Meredith (National Institute for Health Research Collaboration for Applied Health Research and Care (Wessex), Portsmouth Hospitals NHS Trust, Clinical Outcomes Research Group), Alejandra Recio-Saucedo (University of Southampton, Health Sciences, National Institute for Health Research Collaboration for Applied Health Research and Care (Wessex)), David Prytherch (University of Portsmouth, Centre for Healthcare Modelling and Informatics, Portsmouth Hospitals NHS Trust, Clinical Outcomes Research Group), Oliver Redfern (University of Portsmouth, Centre for Healthcare Modelling and Informatics, Nuffield Department of Clinical Neurosciences, University of Oxford), Paul Schmidt (National Institute for Health Research Collaboration for Applied Health Research and Care (Wessex), Portsmouth Hospitals NHS Trust, Clinical Outcomes Research Group), Nicky Sinden (Portsmouth Hospitals NHS Trust, Clinical Outcomes Research Group) and Gary Smith (Bournemouth University, Faculty of Health and Social Sciences).

**Contributors** PG contributed to the design of the study and acquisition of research funding. PG and OCR interpreted the data, and drafted and revised the paper. AM contributed to the design of the study, statistical analysis plan, acquisition of funding and interpretation of data; advised on statistical analysis; contributed to drafting the paper and approved the final manuscript. AR-S and GBS contributed to the interpretation of the results and drafting the paper. Other members of the Missed Care Study Group contributed to the acquisition of funding and/or data, analysis, interpretation of analysis and approval of the paper.

**Funding** This project was funded by the NIHR Health Services and Delivery Research Programme (HS&DR 13/114/17). This paper draws on research and data reported in more detail in the NIHR Journal's Library Publication: P. Griffiths J. Ball, K. Bloor, D. Böhning, J. Briggs, C. Dall'Ora, A. De longh, J. Jones, C. Kovacs, A. Maruotti, P. Meredith, D. Prytherch, A. R. Saucedo, O. Redfern, P. Schmidt, N. Sinden and G. Smith. "Nurse staffing levels, missed vital signs and mortality in hospitals: retrospective longitudinal observational study." Health Services and Delivery Research Journal 2018; 6(38).

**Competing interests** PM, NS and PS are employees of Portsmouth Hospitals NHS Trust (PHT), which had a royalty agreement with The Learning Clinic (TLC) to pay for the use of PHT intellectual property within the Vitalpac product, which expired during the course of this study. DP and GBS are former employees of PHT. PS, and the wives of DP and GS, held shares in TLC until 2015. JB's research has previously received funding from TLC through a Knowledge Transfer Partnership. PG was an unpaid member of the advisory group for NHS Improvement's work developing improvement resources for safe staffing in adult inpatient wards.

**Patient consent for publication** Not required.

**Ethics approval** The study was approved by the National Research Ethics Service, East Midlands – Northampton Committee Ref: 15/EM/0099.

**Provenance and peer review** Not commissioned; externally peer reviewed.

**Data availability statement** No data are available.

**Open access** This is an open access article distributed in accordance with the Creative Commons Attribution 4.0 Unported (CC BY 4.0) license, which permits others to copy, redistribute, remix, transform and build upon this work for any purpose, provided the original work is properly cited, a link to the licence is given, and indication of whether changes were made. See: https://creativecommons.org/licenses/by/4.0/.

**ORCID iDs**
Peter Griffiths http://orcid.org/0000-0003-2439-2857
Gary B Smith http://orcid.org/0000-0003-2070-8455

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
