## [Reviewer comments · BMJ Open]

ARTICLE DETAILS

TITLE (PROVISIONAL)	The association between nurse staffing levels and the timeliness of vital signs monitoring: a retrospective observational study in the UK
AUTHORS	Redfern, Oliver; Griffiths, Peter; Maruotti, Antonello; Recio-Saucedo, Alejandra; Smith, Gary

VERSION 1 – REVIEW

REVIEWER	Emeritus Associate Professor Una Kyriacos University of Cape Town South Africa
REVIEW RETURNED	11-Jul-2019

GENERAL COMMENTS	This is a most interesting and novel study worthy of publication. Well done. It was easier to comment within-text - see attached. Some points to consider in summary: * not all keywords are included on the cover page* I suggest you also add 5.2% for 1,822 ward days (line 5, p 7)* did you mean 2,236 (6.4%) ward days (line 6, p 7)?* Ethical approval - suggest you add the initials of the person/s who removed identifiable information at source (line 49, p8) to differentiate these from researchers who analysed the data* line 17, p9 check if 79% and not 80%* line 21, p9 - does this 16.1% refer to all missed observations? if so, check if this is not 17.1% - so too, check if 31.3% and not 30.1% were delayed?* line 13, p10 - typo (tertiles)* line 55, p10 - first mention of HCA* Conclusion line 28, p12 - levels of registered nurse?* Placement of all table headings on top?* Figure headings below diagrams?* Figure 2 not found* Within all tables - if % indicated as (%) in the heading the symbol % is not needed alongside the Number The reviewer also provided a marked copy with additional comments. Please contact the publisher for full details.
--

REVIEWER	Alexa Doig School of Nursing College of Health and Social Services New Mexico State University USA
REVIEW RETURNED	16-Jul-2019

GENERAL COMMENTS	Excellent, clearly written article. I appreciate the clarity with which the complex modeling was presented both in the methods and results sections. The selection (and definition) of the two outcome variables (missed VS and late VS) are clinically relevant. The presentation of the two models will provide hospital administrators with a clear path towards reducing this aspect of missed nursing care. Page 7, Lines 55-58: NEWS stratification. Is there a validated approach to stratification that can be cited? Other articles I have read use a different stratification scheme (e.g., low = 0-4). Page 10, Lines 1-3: I am not 100% clear what associations are being analyzed in this first sentence. The number of admissions per nurses vs. rate of missed observations + the proportion of higher acute patients and missed observations? Page 10, Lines 5 & 6: Could you elaborate on the meaning or significance of adding a linear interaction term? This may be clear to a statistician, but the primary audience for this article is hospital administrators. I don't think that Table 1 is necessary since all of the data are presented in the text. The table doesn't have a lot of clear organization because the variables are not really related. I also don't think that Figure 1 is necessary. Everything included in the figure was presented clearly in the text.
--

VERSION 1 – AUTHOR RESPONSE

Reviewer: 1

This is a most interesting and novel study worthy of publication. Well done. It was easier to comment within-text - see attached.

- not all keywords are included on the cover page
- I suggest you also add 5.2% for 1,822 ward days (line 5, p 7)
- did you mean 2,236 (6.4%) ward days (line 6, p 7)?
- Ethical approval - suggest you add the initials of the person/s who removed identifiable information at source (line 49, p8) to differentiate these from researchers who analysed the data
- line 17, p9 check if 79% and not 80%
- line 21, p9 - does this 16.1% refer to all missed observations? if so, check if this is not 17.1% - so too, check if 31.3% and not 30.1% were delayed?
- line 13, p10 - typo (tertiles)
- line 55, p10 - first mention of HCA
- Conclusion line 28, p12 - levels of registered nurse?
- Placement of all table headings on top?
- Figure headings below diagrams?
- Figure 2 not found
- Within all tables - if % indicated as (%) in the heading the symbol % is not needed alongside the Number

We are very grateful to the reviewer for their close reading of our manuscript and apologise for these inconsistencies, which we have now corrected.

Reviewer: 2

Excellent, clearly written article. I appreciate the clarity with which the complex modeling was presented both in the methods and results sections. The selection (and definition) of the two outcome variables (missed VS and late VS) are clinically relevant. The presentation of the two models will provide hospital administrators with a clear path towards reducing this aspect is missed nursing care.

Page 7, Lines 55-58: NEWS stratification. Is there a validated approach to stratification that can be cited? Other articles I have read use a different stratification scheme (e.g., low = 0-4).

We thank the reviewer for raising this. Our stratification of the NEWS is based on the acuity groupings described in the study hospital's escalation policy, which we have now included in the supplementary material and referenced in the methods section. To the best of our knowledge, there are no "validated" methods to stratify the NEWS into acuity levels. Current guidance from the Royal College of Physicians (RCP) defines low/medium/high groupings as scores of 1-4, 5-6, 7+ respectively. However, the RCP also suggest that acuity groupings should be agreed locally and are focussed on the need to evoke a graded clinical response. As the frequency (and therefore timeliness) of vital sign observations during the study period was driven by the hospital's protocol, we felt these would be the most appropriate groupings for our analysis.

Page 10, Lines 1-3: I am not 100% clear what associations are being analyzed in this first sentence. The number of admissions per nurses vs. rate of missed observations + the proportion of higher acute patients and missed observations?

We apologise for the confusion here. We have clarified the first two sentences to read:

"Table 3 (Model A) shows the relationship between staffing levels and measure of daily nursing workload with the rate of all missed observations. The rate of missed observations was significantly associated with levels of both RN ($p < 0.0001$) and Nursing Assistant (NA) staffing ($p < 0.001$)."

Page 10, Lines 5 & 6: Could you elaborate on the meaning or significance of adding a linear interaction term? This may be clear to a statistician, but the primary audience for this article is hospital administrators.

The linear interaction term between registered nurses (RNs) and nursing assistants (NAs) was included in the linear model because of the possibility that the effect of one staffing group may be dependent on the effect of the other. For example, if NAs act as labour complements, adding additional registered nurses would be expected to have a greater effect when there are more NAs. We have modified paragraph 4 of the results section to clarify.

I don't think that Table 1 is necessary since all of the data are presented in the text. The table doesn't have a lot of clear organization because the variables are not really related.

We have included Table 1 as the reporting guidelines (RECORD) recommend including a clear description of the study population. Although these data are also presented in the text, we feel that Table 1 provides a useful summary for readers.

I also don't think that Figure 1 is necessary. Everything included in the figure was presented clearly in the text.

We included Figure 1 again to adhere to the RECORD guidance and demonstrate how exclusion criteria were applied. We agree it might be unnecessary within the main manuscript so have moved to the supplementary material.

VERSION 2 – REVIEW

REVIEWER	Emeritus Associate Professor Una Kyriacos University of Cape Town South Africa
REVIEW RETURNED	21-Aug-2019

GENERAL COMMENTS	Thank you for responding to my comments. Just a note: I see that Table 2 includes the symbol % in the heading and following each result - it is not necessary to include the latter.
---

REVIEWER	Alexa Doig New Mexico State University, USA
-----------------	--

REVIEW RETURNED	25-Aug-2019
-------------

GENERAL COMMENTS	All of the questions that I brought up were adequately addressed by the authors. I appreciated the clarity of their explanations. I have no other concerns.
---